# Molecular structures and conformations of protocadherin-15 and its complexes on stereocilia elucidated by cryo-electron tomography

**Johannes Elferich[1,2,3], Sarah Clark[1], Jingpeng Ge[1], April Goehring[1,3], Aya Matsui[1,3], Eric Gouaux[1,3]***

[1]Vollum Institute, Oregon Health and Science University, Portland, United States; [2]RNA Therapeutics Institute, UMass Chan Medical School, Worcester, United States; [3]Howard Hughes Medical Institute, Portland, United States

**Abstract** Mechanosensory transduction (MT), the conversion of mechanical stimuli into electrical signals, underpins hearing and balance and is carried out within hair cells in the inner ear. Hair cells harbor actin-filled stereocilia, arranged in rows of descending heights, where the tips of stereocilia are connected to their taller neighbors by a filament composed of protocadherin 15 (PCDH15) and cadherin 23 (CDH23), deemed the 'tip link.' Tension exerted on the tip link opens an ion channel at the tip of the shorter stereocilia, thus converting mechanical force into an electrical signal. While biochemical and structural studies have provided insights into the molecular composition and structure of isolated portions of the tip link, the architecture, location, and conformational states of intact tip links, on stereocilia, remains unknown. Here, we report in situ cryo-electron microscopy imaging of the tip link in mouse stereocilia. We observe individual PCDH15 molecules at the tip and shaft of stereocilia and determine their stoichiometry, conformational heterogeneity, and their complexes with other filamentous proteins, perhaps including CDH23. The PCDH15 complexes occur in clusters, frequently with more than one copy of PCDH15 at the tip of stereocilia, suggesting that tip links might consist of more than one copy of PCDH15 complexes and, by extension, might include multiple MT complexes.

**\*For correspondence:** gouauxe@ohsu.edu

**Competing interest:** The authors declare that no competing interests exist.

## Editor's evaluation

The authors have done an extremely thorough job of responding to all of the concerns of the reviewers. The present manuscript will be a nice advance in the field of hearing research and is suitable for publication in *eLife*.

## Introduction

Vertebrates sense sound, head movement, and gravity using specialized sensory cells, called hair cells (*McPherson, 2018*). In mammals, hair cells are found in the inner ear and are organized in several specialized organs. Examples include the cochlea, which senses sound, and the utricle, which contributes to movement and gravity sensation (*Ekdale, 2016*). Hair cells harbor sensory microvilli at their apical surface, which are called stereocilia. Stereocilia are rigid, due to being filled with crosslinked actin filaments (*Tilney et al., 1992*), and assemble in a staircase pattern in rows of descending height (*Pickles et al., 1984*). Upon deflection of the stereocilia staircase, tension is exerted on a filament connecting the tip of stereocilia with their taller neighbor, called the tip link (*Pickles et al., 1984*). This

tension then causes the opening of a channel, called the mechanosensory transduction (MT) channel, at the base of the tip link (*Zheng and Holt, 2021*). The molecular mechanism of MT channel function, the conversion from mechanical to electrical signal in the inner ear, remains poorly understood.

The molecular nature of components of the MT machinery has been recently elucidated. The tip link consists of the two-noncanonical cadherins, protocadherin 15 (PCDH15) and cadherin 23 (CDH23) (*Zheng and Holt, 2021*). PCDH15 is situated at the tip of the shorter stereocilia and its two N-terminal cadherin domains bind to the two N-terminal cadherin domains of CDH23 (*Kazmierczak et al., 2007*; *Sotomayor et al., 2012*). The MT channel is likely formed by the transmembrane-channel-like (TMC) protein one or two and the transmembrane protein of the inner ear (TMIE) (*Zheng and Holt, 2021*). PCDH15 assembles in the membrane together with lipoma HMGIC Fusion Partner-Like 5 (LHFPL5, TMHS) (*Ge et al., 2018*, p. 5) and pull-down studies using protein fragments suggest that PCDH15 may also interact with TMC-1 (*Maeda et al., 2014*). PCDH15 is also theorized to interact with the cytoskeleton via whirlin (*Michel et al., 2020*), while TMC may be coupled to the cytoskeleton via CIB2 and ankyrin repeats (*Tang et al., 2020*). However, the molecular composition of the MT machinery and the 3D arrangements of each component remain unknown.

The fundamental transduction activity of the MT machinery is the conversion of force into an electrical signal (*Hudspeth, 1989*). Indeed, the mechanical displacement of the stereocilia can be modeled using Hooke's Law, suggesting the presence of an elastic element, or a 'gating spring,' that couples displacement to the ion channel opening (*Howard and Hudspeth, 1987*). On the one hand, elasticity measurements of PCDH15 suggest that the tip link itself could act as the gating-spring (*Bartsch et al., 2019*), while on the other hand the plasma membrane at the tip of stereocilia, as well as elements that couple the tip link to the cytoskeleton, are other candidates to act as 'elastic elements' (*Powers et al., 2012*). Therefore, insights into the molecular structure and dynamics of the tip link are essential for understanding MT.

The molecular unit of the tip link is proposed to be a heterotetramer of PCDH15 and CDH23, where both cadherins form a parallel dimer (*Dionne et al., 2018*) and come together as a dimer-of-dimers using an anti-parallel 'handshake' of EC-1 and 2 of both cadherins (*Sotomayor et al., 2012*). The extracellular domains of both PCDH15 and CDH23 are a chain of extracellular cadherin (EC) repeats with 11 and 27 repeats, respectively, with a membrane proximal extracellular linker (EL) domain at the C-terminal end (*De-la-Torre et al., 2018*; *Ge et al., 2018*; *Jaiganesh et al., 2018a*). Canonical cadherin repeats form a 'stiff' dimer in the presence of calcium, due to stabilizing calcium-binding sites in the linker regions between cadherin repeats (*Marquis and Hudspeth, 1997*). Both PCDH15 and CDH23 have non-canonical linkers devoid of canonical calcium-binding sites, thus likely promoting conformational mobility (*Araya-Secchi et al., 2016*; *Jaiganesh et al., 2018a*; *Powers et al., 2017*). Notably, the linker between EC9 and EC10 of PCDH15 is flexible in vitro, allowing a bend of up to 90° (*Araya-Secchi et al., 2016*; *Ge et al., 2018*). Bending and extension along this linker have been suggested as underlying the elasticity of the gating spring (*Araya-Secchi et al., 2016*), yet atomic force microscopy measurements and molecular dynamics simulations of PCDH15 also have suggested that individual cadherin domains can unfold to give rise to tip link extension (*Choudhary et al., 2020*; *De-la-Torre et al., 2018*; *Oroz et al., 2019*). However, it is unclear which of these mechanisms occur in situ.

The majority of our current knowledge about the structure of the tip link in situ is based on fixed and stained scanning-electron microscopy (SEM) or freeze-fracture transmission electron microscopy (TEM) of stereocilia (*Kachar et al., 2000*; *Michel et al., 2005*). These images reveal a 120–170 nm long, helically coiled, filament with a diameter of 5 nm and a repeat of 40 nm (*Kachar et al., 2000*). Furthermore, these links appear to bifurcate at the upper and lower insertion sites into 2–3 individual strands, an observation that is difficult to reconcile in light of high-resolution structural data showing that PCDH15 and CDH23 are parallel dimers and that both proteins also harbor membrane-proximal 'dimerization domains' (*De-la-Torre et al., 2018*; *Dionne et al., 2018*; *Ge et al., 2018*). This may be because the imaging methods relied on fixation and staining procedures, which are manipulations that can introduce artifacts or distortions.

More recently, cryo-electron microscopy (EM) imaging has been used to image stereocilia in their native state, avoiding these artifacts (*Metlagel et al., 2019*; *Song et al., 2020*). However, due to rapid damage of the specimen by high-energy electrons, the contrast of this imaging modality is low, making it challenging to visually identify relatively small features, such as the tip link. Therefore,

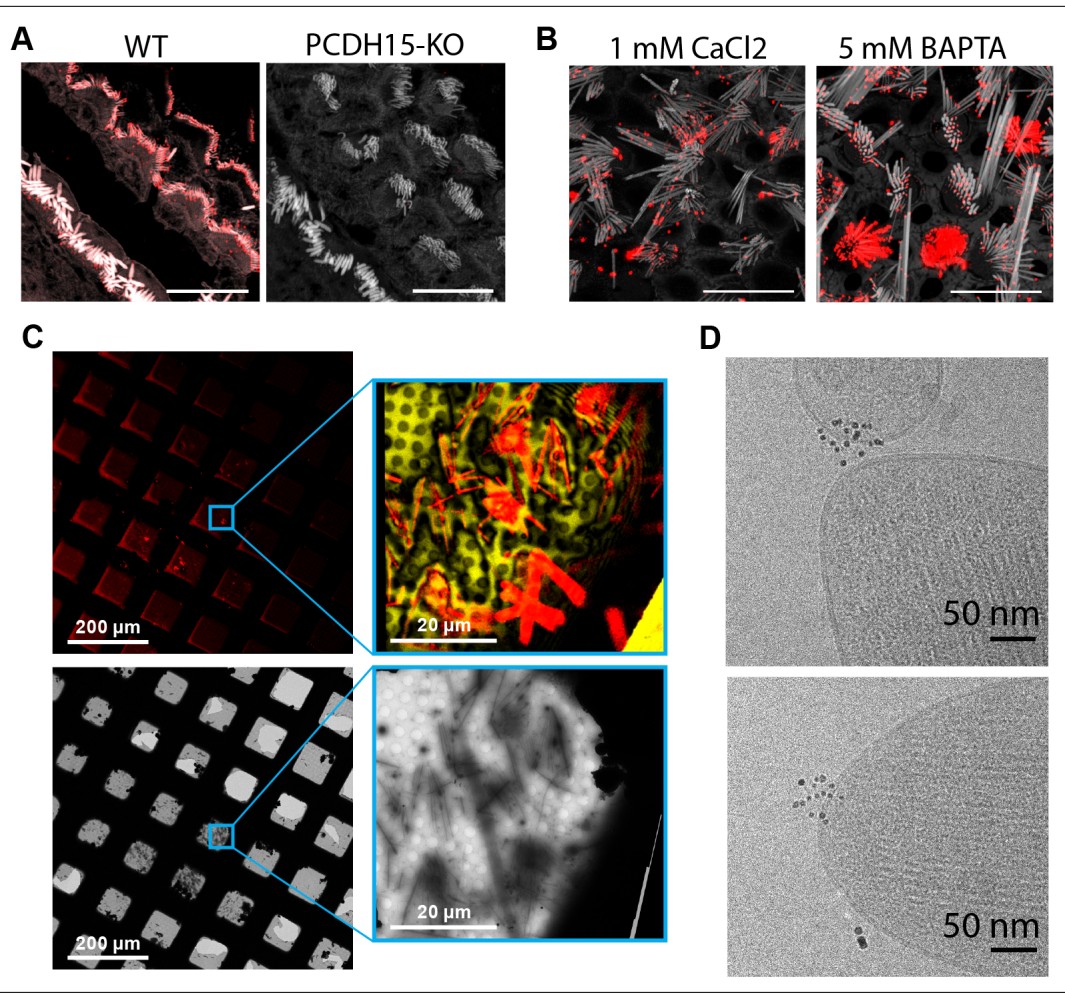

**Figure 1.** Cryo-EM data collection of antibody-stained stereocilia tips supported by cryo-CLEM. (**A**) Immunostaining of WT cochlea or PCDH15 KO with polyclonal antibody raised in rabbits against PCDH15 extracellular domain. Scale bars correspond to 10 µm. (**B**) Immunostaining of WT utricle with polyclonal antibody in media containing 1 mM CaCl$_2$ or with the addition of 5 mM BAPTA. Scale bars correspond to 10 µm. (**C**) CLEM-based screening to identify squares with thin-ice supported stereocilia. The top row shows low- and medium magnification views acquired using a cryo-light microscope. In both views, SiR-actin fluorescence is shown in red. In the medium-magnification view, laser reflection is shown in gold. The bottom rows show the same areas as acquired in a cryo-TEM. (**D**) High-magnification cryo-TEM micrographs of stereocilia tips stained with the anti-PCDH15 rabbit polyclonal antibody, detected with a 5 nm gold coupled secondary antibody. CLEM, correlative light and electron microscopy; WT, wild-type.

labeling approaches are required to unambiguously identify components of the MT machinery. In this study, we combine a stereocilia preparation technique for cryo-EM imaging (*Metlagel et al., 2019*; *Song et al., 2020*) with a highly specific, immune-affinity labeling approach using anti-PCDH15 antibody-coupled gold nanoparticles (AuNP) (*Azubel et al., 2019*) to elucidate the molecular structures and conformational states of tip links under near-native conditions.

## Results

### Cryo-correlative light and electron microscopy imaging of immunolabeled stereocilia

To develop tools to identify PCDH15 in cryo-electron tomograms, we raised polyclonal antibodies (pAbs) against the entire extracellular region of mouse PCDH15. Immunostaining of mouse cochlea using these anti PCDH15 pAbs produces robust staining of stereocilia in hair cells derived from

wild-type (WT) mice that is entirely absent from the stereocilia of PCDH15 knock-out mice, thus demonstrating the specificity and utility of the anti-PCDH15 pAbs (*Figure 1A*). We observed similar staining of stereocilia derived from mouse utricles, both in the presence of calcium and after chelation of calcium using BAPTA, indicating that the antibody binding was independent of calcium and, therefore, independent of PCDH15-CDH23 binding (*Figure 1B*). To facilitate the collection of a large dataset of tomograms focused on the tip region of stereocilia, we adopted a stereocilia preparation technique in which the sensory epithelium is touched to holey carbon support film cryo-EM grids treated with poly-lysine, thus enabling the deposition of stereocilia onto the grid (*Metlagel et al., 2019*). We subsequently identified stereocilia by staining the sample with an actin dye and imaging with cryo-light fluorescence microscopy (*Figure 1C*), allowing us to estimate the number of squares with a favorable number of stereocilia and appropriate ice thickness. We then collected tomograms in these squares, focusing on positions where the tips of stereocilia coincided with holes in the holey carbon film (*Figure 1C*).

In initial experiments, we labeled stereocilia with the anti-PCDH15 pAbs using secondary antibodies conjugated to 5 nm gold particles. In the resulting images, we clearly visualized clusters of 5 nm gold particles in the images of some of the tips (*Figure 1D*). While this validated our immune-labeling approach, we were unable to determine the number of PCDH15 molecules present due to the undefined stoichiometry of both the primary and secondary reagent. Furthermore, the gold particles obstructed the direct observation of the PCDH15 electron density.

## Preparation of a PCDH15 gold-labeling reagent with 1:1 stoichiometry

To specifically label PCDH15 subunits with single gold particles, we developed a high affinity, low off-rate ($k_{off}$=4.6×10⁻⁵ 1 /s), anti-PCDH15 monoclonal antibody (mAb), termed 39G7. By examining the binding of 39G7 to a series of PCDH15 truncation constructs, we determined that the mAb binds to the EC3 cadherin repeat, near the amino terminus (*Figure 2A*). Similar to the anti-PCDH15 pAbs, the 39G7 mAb stained PCDH15 on the surface of vestibular stereocilia, either in the presence or absence of calcium (*Figure 2B*).

To create a reagent to label PCDH15 subunits with a single gold particle, we created a 39G7 Fab construct with a single free cysteine residue at the C terminus of the heavy chain, thus enabling conjugation of the Fab to 3 nm AuNPs (*Azubel et al., 2019*, p. 2). Purification of the 39G7 Fab – AuNP complex via polyacrylamide electrophoresis and size exclusion chromatography (SEC) yielded a homogeneous species in which the Fab was a labeled with a single AuNP (*Figure 2C+D*), devoid of non-AuNP-labeled Fabs. Using this reagent, we reasoned that we could directly count PCDH15 subunits because only a single Fab will bind to each PCDH15 subunit and each Fab is labeled by a single AuNP. We confirmed this prediction by forming a complex between the 39G7 Fab – AuNP and recombinant PCDH15, isolating the complex by SEC (*Figure 2E*) and imaging the complex by single-particle cryo-EM (*Figure 2F*). On micrographs and tomographic reconstructions (*Figure 2—video 1*), we clearly identified pairs of AuNPs bound to the PCDH15 extracellular domain, consistent with a dimeric model of the PCDH15 extracellular domain (*Choudhary et al., 2020*), the 39G7 Fab binding to EC3 and the ability of the Fab – AuNP complex to allow us to identify and count PCDH15 subunits.

## 39G7-AuNP labeling of stereocilia shows that PCDH15 is a dimer

We prepared cryo-EM grids using utricles stained with 39G7 Fab-AuNP and collected tomograms, primarily of stereocilia tips (*Figure 3E*). On approximately half of the tomograms, we identified 3 nm gold particles within ~38 nm of the stereocilia membrane, a distance that is consistent with images of the recombinant PCDH15/39G7 Fab AuNP complex (*Figure 2F*) and with the binding of 39G7 to EC3 of PCDH15 assuming a length of 4.5 nm per cadherin repeat (*Jaiganesh et al., 2018b*). We also identified AuNP labels in tomograms of the stereocilia shaft region, but only in about a third of the tomograms that featured only stereocilia shaft regions. Close inspection of the electron density around the AuNPs frequently revealed ~50 nm long filamentous density consistent with the extracellular domain of PCDH15 (*Figure 3A–B*, *Figure 3-video 1*, *Figure 3-video 2*). In tomograms with relatively thin ice (<200 nm) and an appropriate orientation of the filament to the tilt axis, dimeric features of the PCDH15 domain are evident together with AuNP pairs with a distance of 20 nm, which is consistent with two 7 nm long Fabs and an 8.5 nm helical diameter of the PCDH15 extracellular domain (*Dionne et al., 2018*). This demonstrates that PCDH15 forms a dimer in stereocilia. In some

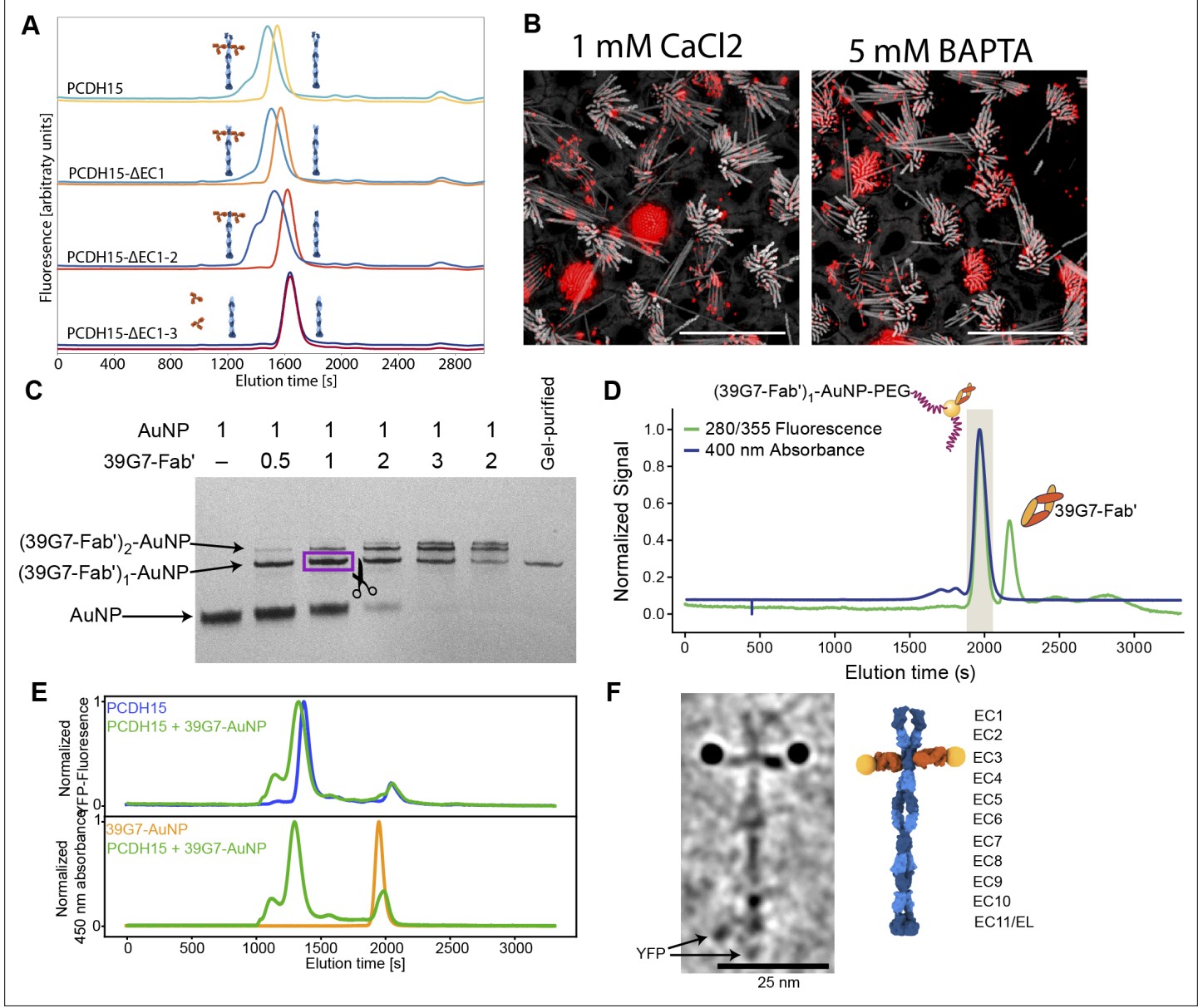

**Figure 2.** Stoichiometric gold staining of PCDH15 using a monoclonal Fab AuNP conjugate. (**A**) FSEC analysis of binding of 39G7 to the PCDH15 extracellular domain. 39G7 addition leads to earlier elution of PCDH15 in full-length constructs and after deletion of EC1 and EC2. Deletion of EC3 abolished 39G7 binding. (**B**) Immunostaining of WT utricle with 39G7 in media containing 1 mM CaCl2 or with the addition of 5 mM BAPTA. The qualitatively similar staining in both conditions suggests that binding can occur in the context of a PCDH15/CDH23 complex (in the presence of calcium) or in the context of unbound PCDH15 (after disruption of tip links by BAPTA). (**C**) PAGE analysis and purification of conjugation between 39G7 Fab' and a 2 nm AuNP. Higher ratios of 39G7-Fab' to AuNP lead to lower amount of free AuNP and higher amounts of AuNP bound to multiple copies of 39G7-Fab'. The fastest migrating 39G7-Fab'-AuNP band was assumed to correspond to a 1:1 complex and purified by cutting it out from the PAGE gel. (**D**) SEC purification of PEG-coated 39G7-Fab'-AuNP-PEG conjugates. By only collecting fractions corresponding to 39G7-Fab'-AuNP-PEG conjugates free Fab', which would disrupt labeling, is removed. (**E**) FSEC analysis of 39G7-Fab'-AuNP-PEG conjugate binding to PCDH15 extracellular domain. Mixing of 39G7-Fab'-AuNP-PEG and PCDH15 extracellular domain leads to faster elution of both molecules at identical times, indicating that they formed a complex. (**F**) Cryo-TEM image of PCDH15 extracellular domain bound to 39G7-Fab'-AuNP-PEG conjugate together with model of the complex. In the model, PCDH15 is shown in blue, 39G7 Fab in orange, and AuNPs as golden spheres.

The online version of this article includes the following video and source data for figure 2:

**Source data 1.** Original image of polyacrylamide gel shown in *Figure 2C*.

**Figure 2—video 1.** Tomogram of recombinant PCDH15 extracellular domain in complex with 39G7-AuNP conjugate.

https://elifesciences.org/articles/74512/figures#fig2video1

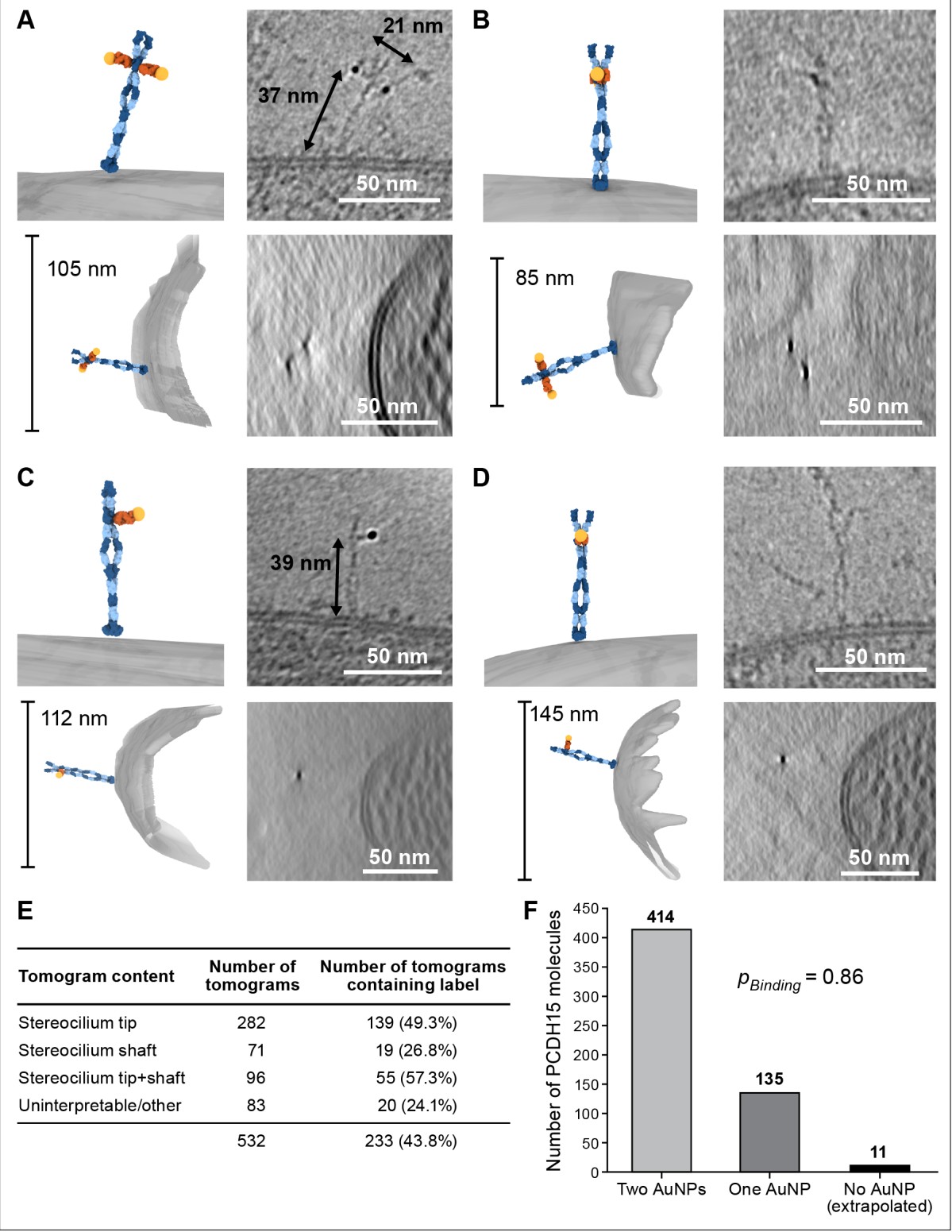

**Figure 3.** 39G7-AuNP conjugate labels PCDH15 dimer in stereocilia. (**A–D**) Representative examples of PCDH15 dimers with two bound AuNPs (**A, B**) or one bound AuNP (**C, D**) image in stereocilia. On the left-hand side of each panel, a manually annotate model is shown in a view roughly corresponding to the electron beam (top) and from the side (bottom). On the right-hand side are corresponding slices through the electron density. Due to the missing-wedge effect, the resolution of the side-view at the bottom is lower, blurring AuNP and precluding direct visualization of PCDH15. For panels (**B**) and (**D**), it is therefore impossible to show AuNPs and PCDH15 in a single projection as the AuNPs are situated above and below PCDH15 in relation to the electron beam. In panels (**C**) and (**D**), there are dimeric features in the PCDH15 density, suggesting that these molecules are PCDH15 dimers

*Figure 3 continued on next page*

Figure 3 continued

with one epitope not bound to 39G7. (**E**) Table detailing numbers and content of collected tomograms. (**F**) Bar chart quantifying the ratio of PCDH15 molecules labeled by one or two AuNPs. Assuming that AuNPs bind independently to the two epitopes in the PCDH15 dimer results suggests that 86% of all epitopes were labeled and only 2% of PCDH15 molecules were unlabeled.

The online version of this article includes the following video for figure 3:

**Figure 3—video 1.** .Tomogram of stereocilium displaying a PCDH15 dimer with two bound 39G7-AuNP conjugates, also depicted in Figure 3A.
https://elifesciences.org/articles/74512/figures#fig3video1

**Figure 3—video 2.** Tomogram of stereocilium displaying a PCDH15 dimer with two bound 39G7-AuNP conjugates, also depicted in Figure 3B.
https://elifesciences.org/articles/74512/figures#fig3video2

**Figure 3—video 3.** Tomogram of stereocilium displaying a PCDH15 dimer with one bound 39G7-AuNP conjugates also depicted in Figure 3C.
https://elifesciences.org/articles/74512/figures#fig3video3

**Figure 3—video 4.** Tomogram of stereocilium displaying a PCDH15 dimer with one bound 39G7-AuNP conjugates, also depicted in Figure 3D.
https://elifesciences.org/articles/74512/figures#fig3video4

cases, we observed individual gold particles instead of a pair of AuNPs (*Figure 3C–D*, *Figure 3-video 3-4*) yet also dimeric features associated with PCDH15 chain, thus indicating that the visualization of individual AuNPs was likely due to incomplete labeling of PCDH15 and not the presence of PCDH15 monomers. We quantified the occurrence of AuNP dimers and monomers (*Figure 3F*) and found a ratio of 3:1. Under the assumption that binding of the 39G7 Fab to either protomer are independent events, we can use this ratio to estimate the fraction of PCDH15 protomers labeled with 39G7-AuNP (0.86) and the fraction of PCDH15 dimers that are completely unlabeled (0.02). While this may be an underestimation of the number of unlabeled PCDH15 molecules, either due to cooperativity of 39G7 binding or due to air-water interface effects, it nevertheless demonstrates that the 39G7 Fab-AuNP robustly labels PCDH15 and therefore allows precise quantification of the number of PCDH15 molecules at the tips of stereocilia.

## Stereocilia tips harbor multiple copies of PCDH15

We carefully quantified the number of PCDH15 molecules in all imaged stereocilia tips (n=396) (*Figure 4F*). Slightly more than half of the stereocilia tips (58%) did not contain a PCDH15 label. While this could be due to an underestimate of the labeling efficiency or due to preferential selection of stereocilia in the tallest row, we believe the most likely explanation is damage to stereocilia tips during the blotting of stereocilia on the cryo-EM grids. In some cases (5.8%), we found PCDH15 molecules in the shaft region just 'below' the stereocilia tips (*Figure 4A*, *Figure 4-video 1*). It is possible that these molecules were initially located on the tip, but diffused away in the time between applying tissue to the grid and plunge-freezing. In 13% of tips, we found a single molecule of PCDH15 at the tip (*Figure 4B+C*, *Figure 4-videos 2 - 3*), similar to commonly depicted models of the MT complex. However, in almost twice the number of stereocilia tips (23%), we found multiple copies of PCDH15 at the tip, either clustered at the tip (*Figure 4D*, *Figure 4-video 4*) or spread across its surface (*Figure 4E*, *Figure 4-video 5*). Because we cannot determine that the maturity of the hair cells from which the stereocilia are derived, multiple PCDH15 molecules at the tips may be due to stereocilia derived from immature hair cells. Indeed, many molecular models of the MT machinery posit the presence of one copy of the PCDH15 dimer bound to a single MT channel. Our data, however, suggest that many tips might harbor multiple copies of the MT machinery, which in turn might underly variation in the ion channel conductance at individual tips (*Beurg et al., 2018*; *Indzhykulian et al., 2013*).

## Tomographic reconstructions of PCDH15 and its complexes

In a few tomograms, we found electron-density extending beyond the tip of PCDH15, consistent with a 120 nm long filament inserted into two distinct membranes (*Figure 5*). While we cannot determine the molecular identity of this density unambiguously, based on the dimensions of this molecule, we speculate that the density corresponds to CDH23. In one example, we found a stereocilium tip with five copies of a PCDH15 dimer, three of which were connected to a possible CDH23 density, which in turn was inserted into a small spherical liposome (*Figure 5A*, *Figure 5—video 1*). Our interpretation of this structure is that it may be a tip link assembly, disrupted during sample preparation, where a portion of the membrane surrounding CDH23 was 'torn off' of the neighboring stereocilium, yielding

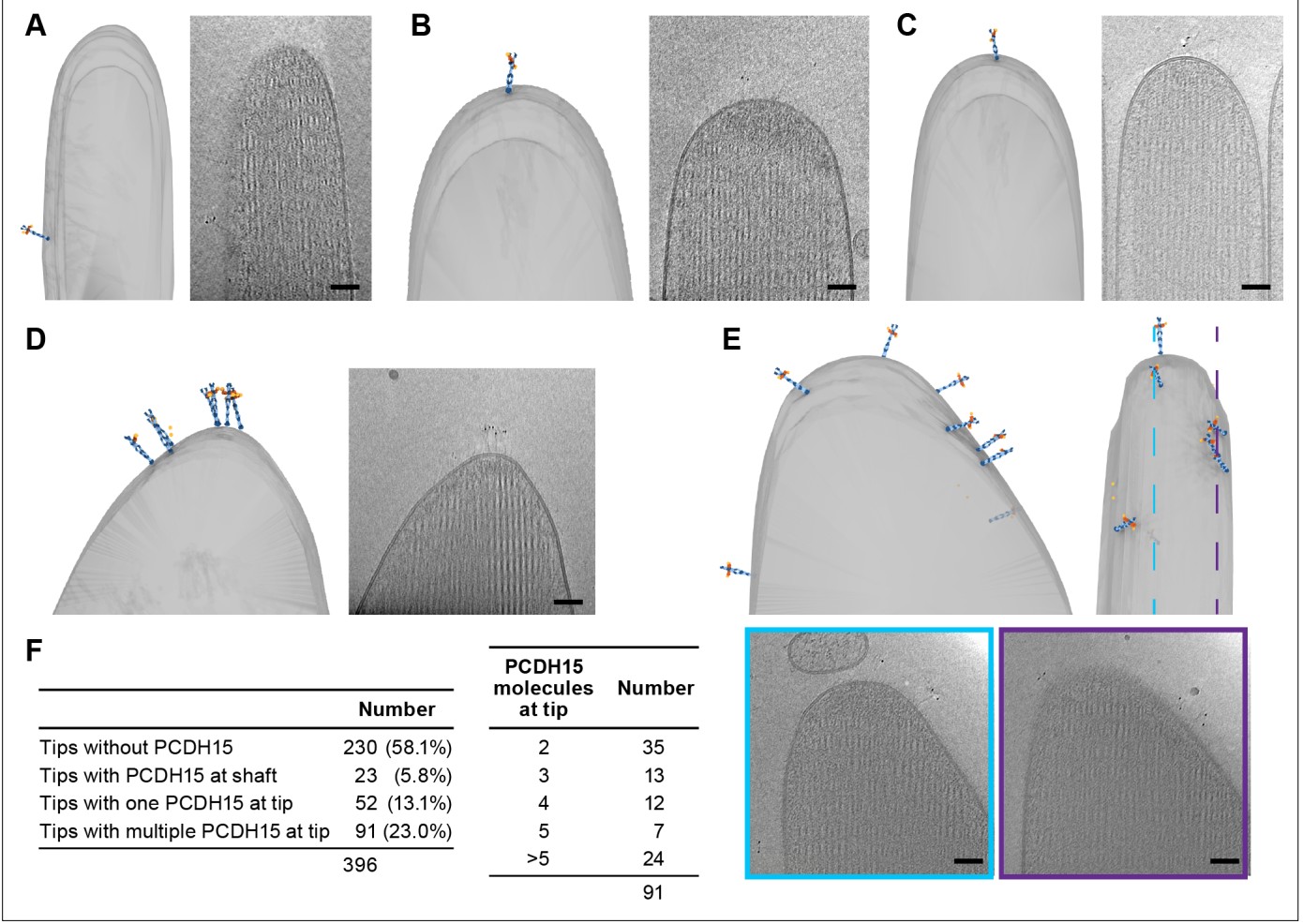

| | Number | | PCDH15 molecules at tip | Number |
|---|---|---|---|---|
| Tips without PCDH15 | 230 (58.1%) | | 2 | 35 |
| Tips with PCDH15 at shaft | 23 (5.8%) | | 3 | 13 |
| Tips with one PCDH15 at tip | 52 (13.1%) | | 4 | 12 |
| Tips with multiple PCDH15 at tip | 91 (23.0%) | | 5 | 7 |
| | 396 | | >5 | 24 |
| | | | | 91 |

**Figure 4.** Stereocilia tips frequently harbor more than one copy of PCDH15. (**A**) Representative example of stereocilia tip with PCDH15 molecule only found in adjacent shaft. (**B, C**) Representative examples of stereocilia tips with one copy of PCDH15 at tip. (**D, E**) Representative examples of stereocilia tips with multiple copies of PCDH15 at tip. (**F**) Table detailing number of imaged stereocilia tips and distribution of PCDH15 molecules found at the tip.

The online version of this article includes the following video for figure 4:

**Figure 4—video 1.** Tomogram of stereocilium tip with a single PCDH15 dimer in the shaft region of the tip, also depicted in Figure 4A.
https://elifesciences.org/articles/74512/figures#fig4video1

**Figure 4—video 2.** Tomogram of stereocilium tip with a single PCDH15 dimer at the apex of the tip, also depicted in Figure 4B.
https://elifesciences.org/articles/74512/figures#fig4video2

**Figure 4—video 3.** Tomogram of stereocilium tip with a single PCDH15 dimer at the apex of the tip, also depicted in Figure 4C.
https://elifesciences.org/articles/74512/figures#fig4video3

**Figure 4—video 4.** Tomogram of stereocilium tip with a cluster of multiple PCDH15 dimers at the apex of the tip, also depicted in Figure 4D.
https://elifesciences.org/articles/74512/figures#fig4video4

**Figure 4—video 5.** Tomogram of stereocilium tip with multiple PCDH15 dimers scattered around the tip, also depicted in Figure 4E.
https://elifesciences.org/articles/74512/figures#fig4video5

CDH23 bound to a liposome. In another case, a cluster of five PCDH15 molecules close to a stereocilium tip is connected to four possible CDH23 densities inserted into a lipid membrane fragment (*Figure 5B*, *Figure 5—video 2*).

We also observed structures in which it appears as though PCDH15, together with a fraction of surrounding membrane, was extracted from the shorter stereocilium (*Figure 5C + D*, *Figure 5—videos 3-4*). In both cases, we found 3–5 copies of PCDH15 and, putatively, CDH23. The fact that we observed most PCDH15 complexes in structures that appear to be the result of partial damage of the tip link is consistent with the earlier interpretation of the missing PCDH15 label is due to damaged

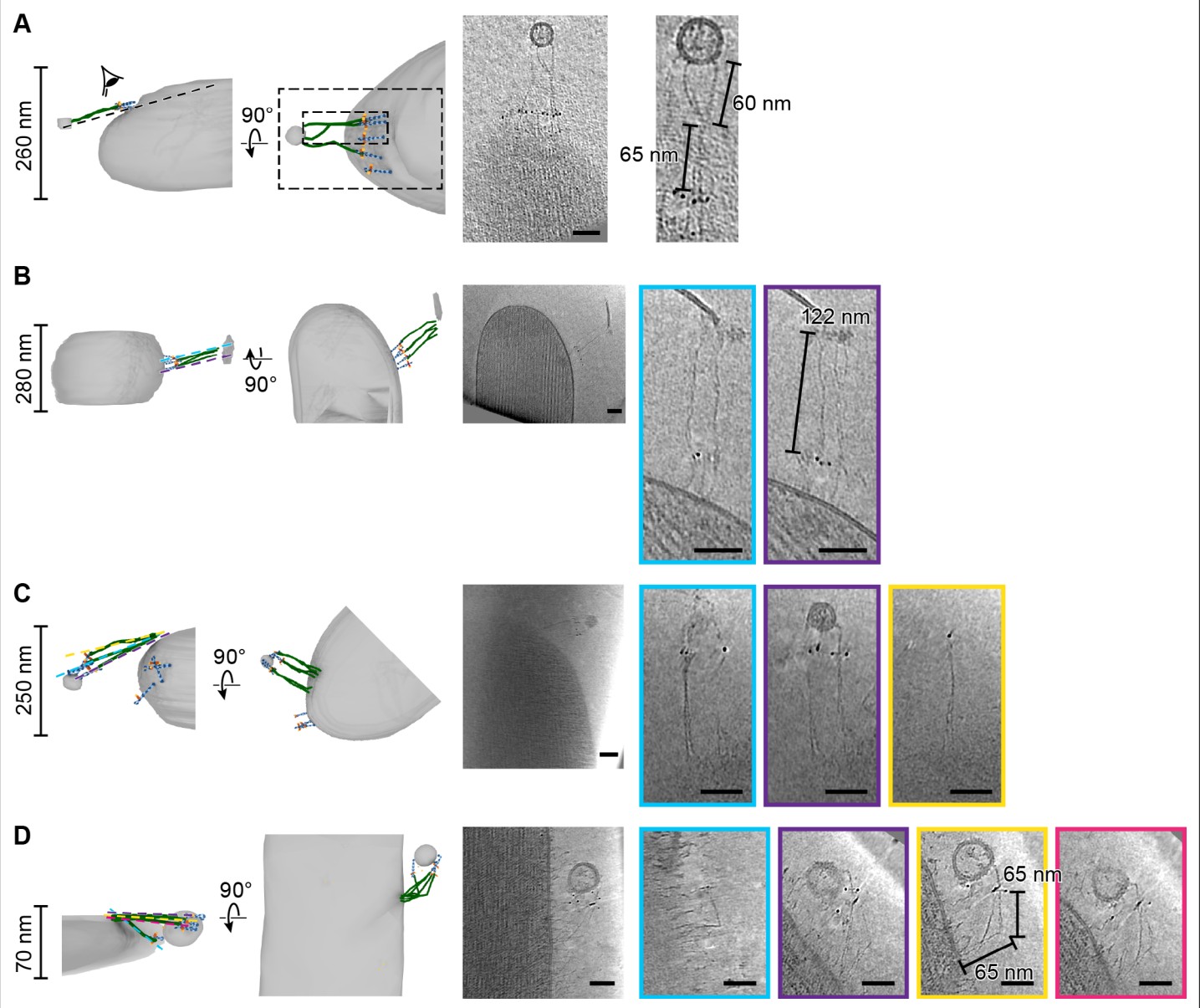

**Figure 5.** PCDH15 complexes are found in clusters. On the left side of each panel, the tomogram annotation is shown from the side and from top. On the sideview ice thickness is indicated. To the right of the annotation is a projection of the cluster density in the context of the stereocilium. Farther to the right are detailed slices of the electron density. The angle and position of each slice are indicated in the sideview of the tomogram annotation. Scale bars correspond to 50 nm. (**A, B**) Representative tomograms showing PCDH15 complexes with PCDH15 at the tip of a stereocilium and the complex partner in a lipid fragment. (**C, D**) Representative tomograms showing PCDH15 complexes with complex partner inserted in the side of a stereocilium and PCDH15 in a lipid fragment.

The online version of this article includes the following video for figure 5:

**Figure 5—video 1.** Tomogram of stereocilium tip containing multiple PCDH15 dimers connected to putative CDH23 filaments in a lipid vesicle, also depicted in Figure 5A.

https://elifesciences.org/articles/74512/figures#fig5video1

**Figure 5—video 2.** Tomogram of stereocilium tip containing multiple PCDH15 dimers connected to putative CDH23 filaments in a lipid membrane fragment, also depicted in Figure 5B.

https://elifesciences.org/articles/74512/figures#fig5video2

**Figure 5—video 3.** Tomogram of a stereocilium tip containing multiple putative CDH23 filaments connected to PCDH15 dimers in a lipid vesicle, also depicted in Figure 5C.

https://elifesciences.org/articles/74512/figures#fig5video3

*Figure 5 continued on next page*

*Figure 5 continued*

**Figure 5—video 4.** Tomogram of a stereocilium shaft containing multiple putative CDH23 filaments connected to PCDH15 dimers in a lipid vesicle, also depicted in Figure 5D.

https://elifesciences.org/articles/74512/figures#fig5video4

tip links. It is also possible that clusters of PCDH15 complexes are more stable than single PCDH15 complexes, and thus more frequently observed.

## A putative tip-link structure surrounded by non-complexed PCDH15 molecules

In one tomogram, we imaged the tip of a stereocilium situated next to a longer stereocilium (*Figure 6A*, *Figure 6-video 1*). While we observed 17 AuNPs at the tip of the shorter stereocilium, we only unambiguously identified the density for four PCDH15 molecules. On one of these PCDH15 molecules, we observed a 120 nm long filament, connecting PCDH15 to the longer stereocilium (*Figure 6B*). We hypothesize that this is most likely an intact tip link. Upon closer inspection (*Figure 6C*), we can clearly identify dimeric features on this PCDH15 molecule. Other adjacent PCDH15 molecules are also clearly dimeric entities (*Figure 6D*), yet are not bound to a potential CDH23 partner. Further down the two stereocilia, we identified another copy of PCDH15, apparently connected via a potential CDH23

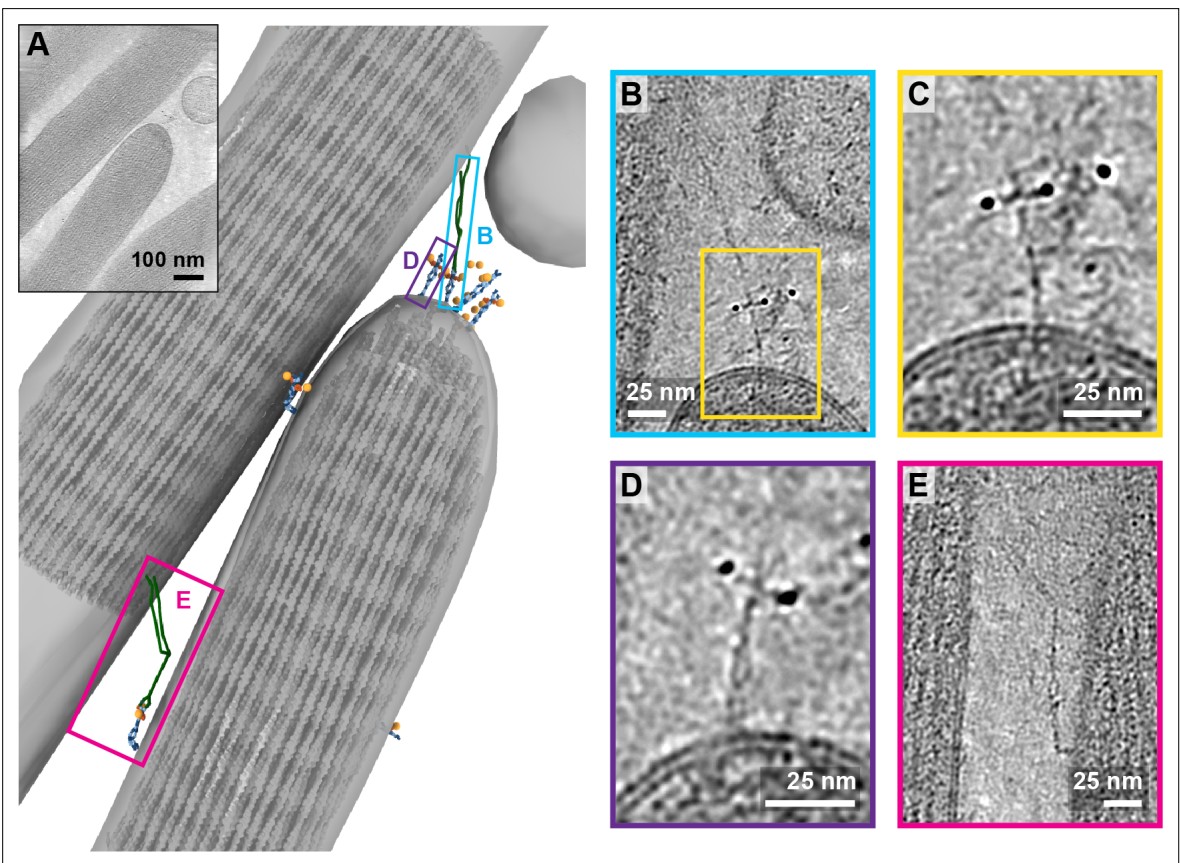

**Figure 6.** Tomogram containing intact tip link. (**A**) Annotation of tomogram showing a putative intact tip link. The inset at the top left shows a projection of the tomogram. (**B**) Close-up view of putative tip link density in the tomogram. (**C**) Close-up of PCDH15 density in tip link. (**D**) Close-up of PCDH15 molecule at tip not bound to a putative CDH23. (**E**) Close-up of PCDH15-complex in the stereocilia shaft region. PCDH15 has a 90° bend near the EC9/EC10 interface.

The online version of this article includes the following video for figure 6:

**Figure 6—video 1.** Tomogram of a stereocilium tip and a stereocilium shaft containing a PCDH15 dimer in the stereocilium tip that is connected to a putative CDH23 filament in the neighboring stereocilium shaft.

https://elifesciences.org/articles/74512/figures#fig6video1

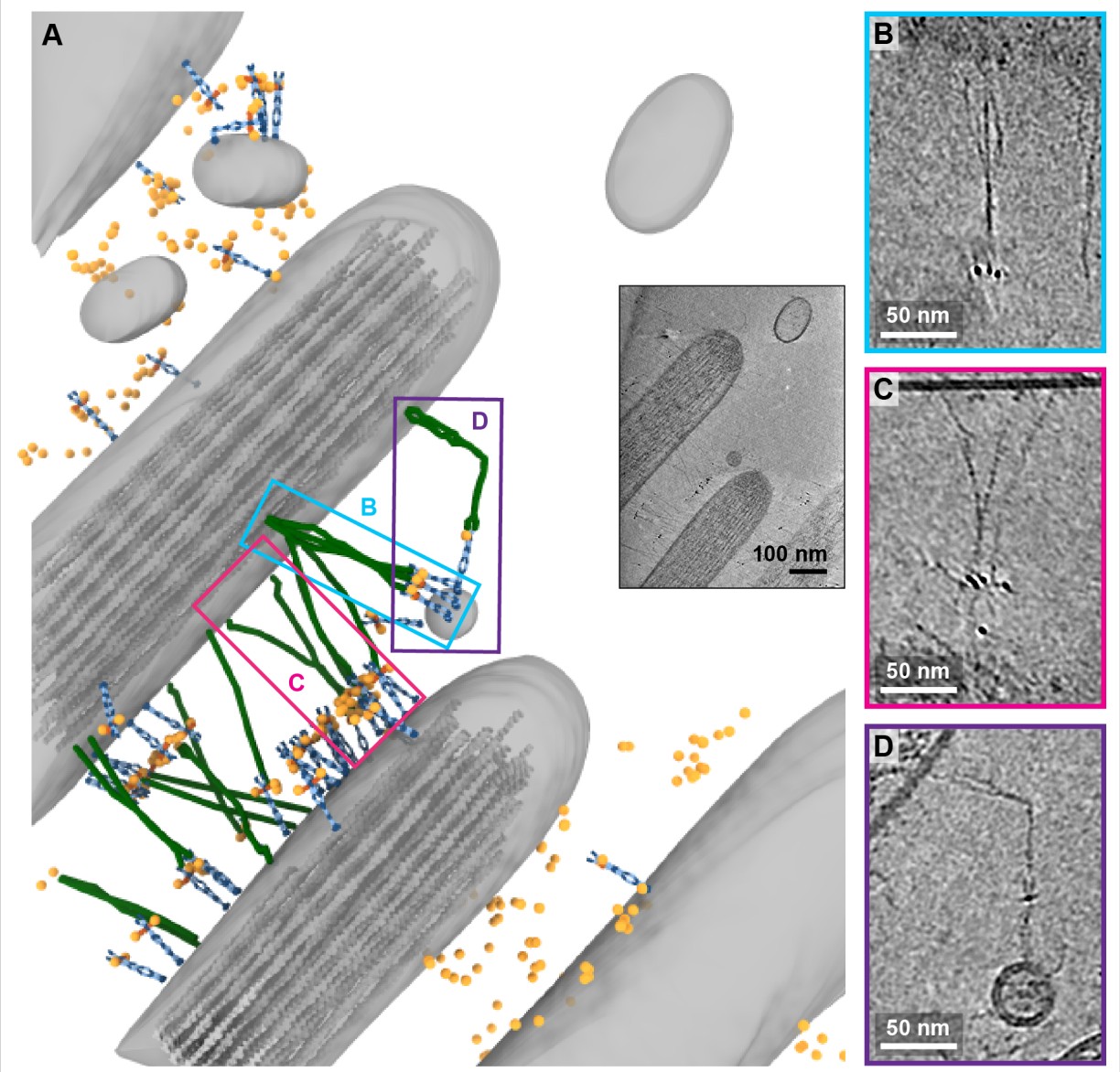

**Figure 7.** Tomogram depicting lateral links containing PCDH15 in small stereocilia. (**A**) Annotation of tomogram showing developing stereocilia including PCDH15-containing 'lateral links.' The inset at the bottom right shows a projection of the tomogram. (**B, C**) Close-up of putative CDH23 molecules clustering together into single strand. (**D**) Close-up of a putative CDH23 molecule with distinct bend halfway between N- and C-terminal end.

The online version of this article includes the following video for figure 7:

**Figure 7—video 1.** Tomogram of stereocilia connected by tens of PCDH15-containing complexes.
https://elifesciences.org/articles/74512/figures#fig7video1

molecule to the other stereocilium (*Figure 6E*). In this case, PCDH15 appears to be bent, by about 90°, near the juncture of the EC9-EC10 cadherin domains, a 'joint' between cadherin domains that has previously been identified as flexible in cryo-EM and crystallography structures of PCDH15 extracellular domain constructs (*Araya-Secchi et al., 2016*; *Ge et al., 2018*). We speculate that this link may be in the process of being trafficked to the tip and that the EC9-EC10 bend helps to accommodate the long filament within the relatively narrow space between stereocilia.

## Lateral links consisting of PCDH15 complexes

While in the majority of tomograms, we identified 10 or fewer copies of PCDH15, several tomograms exhibit numbers of labels consistent with hundreds of copies of PCDH15. In the most striking example

(*Figure 7A*, *Figure 7-video 1*), we observed these labels between stereocilia that appear thinner and less densely packed with actin than other stereocilia. In this tomogram, we counted 309 AuNPs and modeled 56 PCDH15 molecules in cases where we observed the corresponding electron density. Many of the PCDH15 molecules are involved in links to neighboring stereocilia, via a 120 nm filament, which again may be a CDH23 molecule. We observed density for 20 putative CDH23 molecules, molecules that likely participate in the 'lateral links' which form early in development of stereocilia bundles and are composed of PCDH15 and CDH23 (*Michel et al., 2005*). Inspection of the densities shows that multiple copies of the potential CDH23 molecules cluster together, primarily via their N-terminal half so as to obscure identification of individual stands (*Figure 7B+C*). We speculate that these clusters might have given rise to the appearance of single tip links with multiple upper and lower insertion points in stained EM images. Furthermore, we find examples of putative CDH23 molecules that appear bent halfway between the N- and C-termini, indicating that CDH23 might also have a flexible cadherin domain 'joint,' similar to the EC9-EC10 interface of PCDH15 (*Figure 7D*). A potential candidate for this joint is the non-canonical linker between EC12 and EC13 of CDH23 (*Jaiganesh et al., 2018a*).

## Discussion

The combination of cryo-EM and AuNP labeling provides a molecular resolution view of the structure, stoichiometry, and organization of PCDH15 and its complexes on the stereocilia of hair cells in a near-native environment. This work is the first look at PCDH15 and associated protein and lipid membrane complexes in an unfixed and unstained environment, unambiguously demonstrating the presence of PCDH15 as a dimer (*Indzhykulian et al., 2013*; *Kachar et al., 2000*; *Michel et al., 2005*).

We observe clusters of PCDH15 in complexes with other filamentous proteins, perhaps including CDH23, on the stereocilium tip and shaft, as well as multiple copies of PCDH15 at stereocilium tips, ranging from 2 to more than 5. Because the majority of the PCDH15 molecules we observed were not bound to other filamentous proteins and therefore did not form tip links, the mechanistic underpinnings of multiple PCDH15 molecules, if any, are unclear. One possibility is that each PCDH15 molecule was coupled to a functional MT channel and the tip link was ruptured during cryo-EM grid preparation. This would imply that there are multiple MT complexes per stereocilium tip, as suggested previously (*Beurg et al., 2018*). Confocal microscopy images of stereocilia expressing mCherry-tagged TMC1 indicate that there are an average of 7.1 TMC1 channels per stereocilium tip derived from the inner hair cells of P4 aged mice. The number of TMC1 per stereocilium tip varies from four to twenty depending on the cochlear localization, mirroring the high degree of variability we observed for PCDH15.

It is also possible that a large number of PCDH15 molecules may be necessary for rebinding to CDH23 in the event of tip link breakage. Atomic force microscopy experiments indicate that the lifetime of the PCDH15-CDH23 bond is only ~8 s at resting tension, suggesting that the tip link is a highly dynamic connection (*Mulhall et al., 2021*). A pool of nearby PCDH15 molecules may enable fast recovery after tip links are broken. It has also been suggested that intermediate PCDH15-PCDH15 tips form first during tip link regeneration, followed by mature PCDH15-CDH23 tip links, necessitating the presence of multiple PCDH15 molecules (*Indzhykulian et al., 2013*).

Our tomograms also revealed distinct structural features of PCDH15 and putative CDH23 molecules. Most intact complexes were approximately straight, ~120 nm length filaments consisting of a PCDH15 dimer joined to a putative CDH23 dimer. However, in several of these filamentous assemblies, we observed a ~90° bend in PCDH15 or the complex partner. PCDH15 is bent at approximately the EC9-EC10 interface, which has been identified as flexible in cryo-EM and crystal structures of recombinant PCDH15, while the putative CDH23 entity appears to be bent near EC13. Crystal structures of CDH23 identified non-canonical linker regions and altered $Ca^{2+}$ binding sites within this region, motifs that could confer increased flexibility (*Jaiganesh et al., 2018a*). Furthermore, we often observe a 'splitting' of the possible CDH23 dimer at various points, most frequently at the C-terminal end. This type of structural heterogeneity is not observed for PCDH15. Crystallography studies of CDH23 have noted a curious lack of multimerization interfaces between CDH23 protomers, consistent with our observations, yet a rationale for minimal interchain CDH23 interactions remains to be determined (*Jaiganesh et al., 2018b*), other than to reduce likelihood of CDH23 aggregation.

In selected stereocilia, we note that the high densities of lateral PCDH15-containing filaments are consistent with our capture of an immature hair bundle. While the utricles employed in our study were isolated from P6 to P9 mice, lateral links are still present until approximately P9 (*Goodyear et al., 2005*). These extensive lateral links stabilize the stereocilia during the early stages of development by acting as cohesive tethers, gradually being pruned until only the tip link remains in a mature hair bundle (*Boëda et al., 2002*; *Michel et al., 2005*).

Our work highlights the power of gold immunolabeling and cryo-EM to study rare protein complexes. The MT complex is notorious for its low abundance, conspiring to make studies of the mechanism of MT channel gating by tip-link tension challenging. The images presented here reveal structural features of the MT machinery and demonstrate that the employed techniques can be used to visualize single molecules in their native environment. It will be exciting to use this technique to explore additional components of the MT complex, including TMC1 and TMIE, in order to define their locations, stoichiometries, and structures within the architecture of the MT machinery.

# Materials and methods

**Key resources table**

| Reagent type (species) or resource | Designation | Source or reference | Identifiers | Additional information |
|---|---|---|---|---|
| Gene (*Mus musculus*) | *Pcdh15* | Uniprot | Q99PJ1 | |
| Cell line (*Spodoptera frugiperda*) | Sf9 | Thermo Fisher Scientific | 12659017 RRID:CVCL_0549 | |
| Cell line (*Homo sapiens*) | HEK293 tsa 201 | ATCC | CRL-11268 RRID:CVCL_1926 | |
| Recombinant DNA reagent | PCDH15 EC1-EL-YFP expression plasmid | This paper | | Created using pEG BacMam (Addgene plasmid # 160451) |
| Recombinant DNA reagent | 39G7 Fab' expression plasmid | This paper (created by VectorBuilder) | | Backbone described in d oi:10/fhwrn3 |
| Antibody | Anti-PCDH15 (Rabbit polyclonal) | This paper (created by Genscript) | | IF: 10 µg/ml |
| Antibody | Anti-PCDH15 39G7 (Rabbit monoclonal) | This paper (created by Genscript) | | IF: 10 µg/ml |
| Biological sample (*M. musculus*) | Inner ear | | | P6–P9 mice |
| Chemical compound, drug | 10 nm gold fiducials | Ted Pella | 90010 | |
| Chemical compound, drug | HAuCL$_4$ | Sigma-Aldrich | 520918 | |
| Chemical compound, drug | 3-MBA | Sigma-Aldrich | 451436 | |
| Chemical compound, drug | mPEG-550-SH | Creative PEGworks | PLS-607 | |
| Software, algorithm | SerialEM | 10.1016 /j.jsb.2005.07.007 | *RRID*:SCR_017293 | |
| Software, algorithm | IMOD | doi.org.10.1006/jsbi.1996.0013 | *RRID*:SCR_003297 | |
| Software, algorithm | Topaz | doi.org.10.1038/s41467-020-18952-1 | | |
| Software, algorithm | TomoAlign | doi.org.10.1016/j.jsb.2019.01.005 | | |
| Software, algorithm | Motioncor2 | doi.org.10.1038/nmeth.4193 | *RRID*:SCR_016499 | |
| Software, algorithm | ChimeraX | doi.org.10.1002/pro.3943 | RRID:SCR_015872 | |

## Cell lines

Sf9 cells (Thermo Fisher Scientific 12659017) were cultured in sf-900 III SFM medium at 27°C. HEK293 tsa201 cells (ATCC CRL- 11268) were cultured in suspension in Freestyle 293 expression medium supplemented with 1% (v/v) fetal bovine serum at 37°C. Cells are routinely tested for mycoplasma

contamination using CELLshipper Mycoplasma Detection Kit M-100 from Bionique. All of our cells are mycoplasma-free. We have not used any cell lines from the list of commonly misidentified cell lines.

## PCDH15 EC1-EL expression and purification

A gene encoding the amino acid sequence of the mouse PCDH15 extracellular region (Uniprot entry Q99PJ1), from the first cadherin domain (EC1) to the membrane-proximal 'EL' domain (PCDH15 EC1-EL), was synthesized and cloned into a pBacMam vector (*Goehring et al., 2014*), and included a C-terminal yellow fluorescent protein (YFP) fluorophore followed by a polyhistidine tag. This construct was used to generate baculovirus, which was then employed to infect HEK293 cells as previously described (*Goehring et al., 2014*). Approximately 96 hr after viral transduction, the cell medium was harvested and the secreted PCDH15 EC1-EL protein was isolated by metal ion affinity chromatography, and further purified by SEC in TBS Buffer (20 mM Tris pH8, 150 mM NaCl). The final material was concentrated to ~2 mg/ml, aliquoted and stored at –80°C.

## Antibody generation

Rabbit polyclonal and monoclonal antibodies were generated using standard techniques by Genscript using the soluble PCDH15 EC1-EL extracellular region as the antigen. Polyclonal serum was used to isolate antibodies using affinity-purification with the PCDH15 EC1-EL extracellular domain. Hybridoma supernatants were screened against the PCDH15 EC1-EL antigen in the presence and absence of 1 mM calcium to identify clones that recognize the calcium-bound and apo forms of the protein. Supernatants that tested positive by ELISA were further screened by fluorescence detection chromatography (FSEC) and Western blot, and the clone 39G7 was sequenced and monoclonal antibody was produced recombinantly.

## Immunofluorescence

Cochleas and utricles were dissected from mice at ages P6–P9 in DMEM/F12 media (Gibco). The tissue was incubated for 30 min in 10 µg/ml of indicated antibody in DMEM/F12 medium, followed by washing, three times for 10 min each, in the same medium. Where indicated, 5 mM BAPTA was included in staining and washing media to chelate calcium. The tissue was next fixed in a buffer composed of 4% paraformaldehyde in phosphate-buffered saline (PBS) for 10 min. After three washes in PBS, the tissue was permeabilized and blocked using PBS with 0.1% Triton X-100, 5% bovine serum albumin (BSA), and 10% standard goat serum. Subsequently, the tissue was stained with goat-anti-rabbit antibodies fused to Alexa-594 and phalloidin fused to Alexa-405. After three washes with PBS, the tissue was mounted with Vectashield mounting media and imaged using a Zeiss LSM 980 confocal microscope using a 63×/1.49 NA objective.

## AuNP generation

A solution of 84 mM 3-mercaptobenzoic acid (3-MBA) in methanol was mixed with a 28 mM solution of $HAuCl_4$ in methanol at 7:1 molar ratio, followed by 2.5 volumes of water. The pH was adjusted by adding concentrated aqueous NaOH to a final concentration of 100 mM NaOH. This solution was mixed by end-over-end rotation for at least 16 hr. Afterward, the solution was diluted with 27% methanol to achieve a final concentration of 2.5 mM 3-MBA. $NaBH_4$ was added to a final concentration of 2 mM using a fresh 150 mM stock solution prepared in 10 mM NaOH. After mixing for 4.5 hr, the gold particles were precipitated by adjusting the NaCl concentration to 100 mM and by adding methanol to a final concentration of 70%. Gold AuNPs were pelleted by centrifugation at 5000 rpm for 20 min and washed with 70% methanol. The pellet was dried overnight in a desiccator and re-suspended in water.

An expression construct for a Fab' fragment of 39G7 was designed using a dual-promoter Sf9-expression plasmid (Vectorbuilder). For the 39G7 light chain, we replaced the native signal peptide with the GP64 signal peptide and inserted the coding sequence downstream of the PH promoter. For the 39G7 heavy chain, we replaced the native signal peptide with the GP64 signal peptide and truncated the coding sequence after G256, thereby removing the Fc fragment but retaining C243 and C248 at the C-terminus. The coding sequenced was then inserted after the P10 promoter and fused with a histidine tag at the C-terminus. The 39G7 Fab' was expressed in Sf9 cells for 96 hr at 27°C. The media was adjusted to pH 8, cleared by centrifugation and then concentrated to about 100 ml using

a tangential-flow concentrator. The concentrated media was pumped over a 5 ml metal ion affinity column, equilibrated with PBS. The column was washed extensively with PBS supplemented with 30 mM imidazole and Fab' was eluted with PBS supplemented with 500 mM imidazole. The yield was 2 mg of Fab' per liter of culture. The pooled fractions were concentrated to 5 mg/ml and aliquots were plunge-frozen in liquid nitrogen and stored at –80°C.

To conjugate the Fab to the AuNPs, an aliquot of 39G7 Fab' was thawed and incubated with 2 mM tris (2-carboxyethyl) phosphine hydrochloride (TCEP) for 1 hr at 37°C. After clarification by ultracentrifugation, the 39G7 Fab was applied to an SEC column equilibrated with TBE buffer (100 mM Tris, 100 mM boric acid, and 2 mM EDTA). The peak fractions were concentrated immediately to 2 mg/ml and a test conjugation was set up at different molar ratios of Fab' and AuNP (4:1, 2:1, 1:1, 1:2, and 1:4). After a 30-min incubation at 37°C, the reactions were analyzed on a 12.5% PAGE gel made with TBE and 10% glycerol. The condition with the highest yield of the 1:1 AuNP/Fab complex was chosen for large-scale conjugation. To coat the AuNPs with PEG, the conjugate was immediately separated on a PAGE gel following and 1:1 39G7:AuNP recovered in TB buffer. After concentration, 1 mM of mPEG550-SH was added and incubated for 60 min at 37°C. The conjugate was then purified by SEC using a Superose 6 column equilibrated with PBS. Fractions corresponding to 39G7-AuNP conjugate were pooled, concentrated to 1 µM (assuming an extinction coefficient of $3\times10^6$ M$^{-1}$ cm$^{-1}$ at 500 nm), and stored at 4°C.

## Grid preparation

Utricles were dissected from P8 to P11 mice in DMEM/F12 buffer. Otoconia were removed using an eyelash. Utricles were incubated for 30 min in DME M/F12 containing 100 nM 39G7-AuNP and 500 nM SiR-actin and then washed three times for 5 min in DMEM/F12. Utricles were then placed stereocilia-side down on C-Flat 200 mesh copper grids with a 2/1 spacing carbon film that were pretreated with 0.1 mg/ml poly-D-lysine and suspended in 20 µl drops of DMEM/F12. After 2–3 s, the utricle was removed and placed on another region of the grid for up to three times. The grid was removed 'edge-first' from the drop and excess liquid was removed by touching the edge to a piece of Whatman No. 40 filter paper. A 2.5 µl drop of DMEM/F12 containing 0.05% fluorinated octyl-maltoside and 10 nm highly uniform gold fiducials at an OD$_{500}$ of roughly 5.0 were added to the grid. The grid was then placed on a manual blotting apparatus and excess solution was removed by placing a 595 filter paper (Ted Pella) for 6–10 s to the side of the grid without sample. Afterward, the grid was rapidly plunged into a mixture of ethane and propane cooled to liquid nitrogen temperature. Grids were imaged in a CMS196 cryostage (Linkam) on an LSM880 confocal microscope (Zeiss) using the AiryScan detector. Grids that did not exhibit SiR-actin fluorescence in multiple squares or had excessive damage to the carbon support were discarded.

## Tilt series acquisition

Tilt series of the recombinant PCDH15 EC1-EL extracellular region labeled with the 39G7-AuNP was acquired using a Thermo Fisher Arctica microscope operated at 200 keV, without an energy filter, on a Gatan K3 detector. Tilt series of stereocilia were obtained on a Thermo Fisher Krios microscope operated at 300 keV with an energy filter on a Gatan K3 detector. A subset of tomograms was obtained from a Krios microscope also equipped with a spherical aberration corrector.

In all cases, tilt series were obtained from –60° to 60° using a 3° interval using SerialEM. Data were either acquired by sweeping from –30° to 60° and then from –30° to –60° or by using a grouped dose symmetric scheme (*Hagen et al., 2017*). In all cases, the total electron dose was 80–120 e−/ A$^2$. Defocus was varied between –2.5 and –4.0 µm. Movies were motion corrected using MotionCor2 (*Zheng et al., 2017*).

## Tomography processing

Tomograms were reconstructed using the IMOD program (*Mastronarde and Held, 2017*), employing the 10 nm gold fiducials for alignment. In some cases, the 3 nm AuNP label was also included as a fiducial. The tilt series was binned by 4 to a final pixel size of 6.6 A, CTF corrected, the 10 nm gold particles were subtracted, and the tilt series was dose-weighted. Tomograms were reconstructed using a SIRT-like filter with eight iterations. In cases where a substantial amount of sample deformation was observed, tomograms were reconstructed using the TomoAlign program (*Fernandez et al., 2019*),

using polynomials and the 'thick' preset. If indicated, tomograms were denoised using the denoise3d model of Topaz (*Bepler et al., 2020*).

### Tomogram annotation

Membranes were annotated manually in XY slices using the 3dmod program (*Kremer et al., 1996*). Actin filaments were manually annotated in 26.4 nm thick slices perpendicular to the stereocilium length. AuNP nanoparticles were either manually annotated or annotated using the findbeads3d program of the IMOD suite. PCDH15 was positioned by identifying the membrane insertion point and the PCDH15 tip in the 'Slicer' windows of the 3dmod program. The density of the PCDH15 model was then aligned along these two points and, in some cases, was rotated until the projection matched the density visible in the slicer window. A low-pass filtered electron density of actin was placed along the annotated filaments using the clonevolume program of the IMOD suite. The final render of each model was performed using USCF ChimeraX (*Pettersen et al., 2021*).

## Acknowledgements

The authors would like to thank Janette Myers, Craig Yoshioka, and Claudia Lopez at the PNCC at OHSU and Xiaowei Zhao and Shixin Yang at the Janelia Cryo-Electron Microscopy facility for help with data collection. The authors also would like to thank Lauren-Ann Metskas, Eileen O'Toole, and Songye Chen for instructions for tomography data acquisition and data processing. The authors would like to thank Maia Azubel for helpful discussions about AuNP synthesis and conjugation. Furthermore, the authors would like to thank Lori Vaskalis for assistance in preparing figures and Rashell Hallford for careful editing of the manuscript. EG is the Bernard and Jennifer LaCroute Chair in Neuroscience and an investigator with the Howard Hughes Medical Institute. A portion of this research was supported by NIH grant U24GM129547 and performed at the PNCC at OHSU and accessed through EMSL (grid.436923.9), a DOE Office of Science User Facility sponsored by the Office of Biological and Environmental Research.

## Additional information

### Funding

| Funder | Grant reference number | Author |
| --- | --- | --- |
| National Institute of General Medical Sciences | U24GM129547 | Eric Gouaux |

The funders had no role in study design, data collection and interpretation, or the decision to submit the work for publication.

### Author contributions

Johannes Elferich, Conceptualization, Data curation, Formal analysis, Investigation, Visualization, Writing – original draft, Writing – review and editing; Sarah Clark, Formal analysis, Investigation, Writing – original draft, Writing – review and editing; Jingpeng Ge, Conceptualization, Investigation, Writing – review and editing; April Goehring, Formal analysis, Investigation, Writing – review and editing; Aya Matsui, Investigation, Writing – review and editing; Eric Gouaux, Conceptualization, Funding acquisition, Supervision, Writing – review and editing

### Author ORCIDs

Johannes Elferich http://orcid.org/0000-0002-9911-706X
Jingpeng Ge http://orcid.org/0000-0001-6164-1221
April Goehring http://orcid.org/0000-0002-2592-6956
Eric Gouaux http://orcid.org/0000-0002-8549-2360

### Ethics

This study was performed in strict accordance with the recommendations in the Guide for the Care and Use of Laboratory Animals of the National Institutes of Health. All of the animals were handled according to approved institutional animal care and use committee (IACUC) protocols

(TR01_IP00000905) of Oregon Health and Science University. Mice were euthanized according to the American Veterinary Medical Association Guidelines for Euthanasia of Animals before samples were prepared.

## Decision letter and Author response

Decision letter https://doi.org/10.7554/eLife.74512.sa1
Author response https://doi.org/10.7554/eLife.74512.sa2

## Additional files

### Supplementary files

• Transparent reporting form

### Data availability

Tomograms depicted in the figures have been deposited in the EMDB under accession codes EMD-25046, EMD-25047, EMD-25048, EMD-25049, EMD-25050, EMD-25051, EMD-25052, EMD-25053, EMD-25054, EMD-25055, EMD-25056, EMD-25057, EMD-25058, EMD-25059, EMD-25060, and EMD-25061 (in order of appearance). The corresponding tilt-series have been deposited in EMPIAR under accession code EMPIAR-10820. The tilt series of tomograms not depicted in this paper have been deposited in EMPIAR under accession code EMPIAR-10898.

The following dataset was generated:

| Author(s) | Year | Dataset title | Dataset URL | Database and Identifier |
|---|---|---|---|---|
| Elferich J, Clark S, Ge J, Goehring A, Matsui A, Gouaux E | 2021 | Tomogram of mPCDH15/39G7-AuNP complex | https://www.ebi.ac.uk/emdb/EMD-25046 | Electron Microscopy Data Bank, EMD-25046 |
| Elferich J, Clark S, Ge J, Goehring A, Matsui A, Gouaux E | 2021 | Tomogram of mouse stereocilia containing PCDH15 molecules labeled with 39G7-AuNPs | https://www.ebi.ac.uk/emdb/EMD-25047 | Electron Microscopy Data Bank, EMD-25047 |
| Elferich J, Clark S, Ge J, Goehring A, Matsui A, Gouaux E | 2021 | Tomogram of mouse stereocilia containing PCDH15 molecules labeled with 39G7-AuNPs | https://www.ebi.ac.uk/emdb/EMD-25048 | Electron Microscopy Data Bank, EMD-25048 |
| Elferich J, Clark S, Ge J, Goehring A, Matsui A, Gouaux E | 2021 | Tomogram of mouse stereocilia containing PCDH15 molecules labeled with 39G7-AuNPs | https://www.ebi.ac.uk/emdb/EMD-25049 | Electron Microscopy Data Bank, EMD-25049 |
| Elferich J, Clark S, Ge J, Goehring A, Matsui A, Gouaux E | 2021 | Tomogram of mouse stereocilia containing PCDH15 molecules labeled with 39G7-AuNPs | https://www.ebi.ac.uk/emdb/EMD-25050 | Electron Microscopy Data Bank, EMD-25050 |
| Elferich J, Clark S, Ge J, Goehring A, Matsui A, Gouaux E | 2021 | Tomogram of mouse stereocilia containing PCDH15 molecules labeled with 39G7-AuNPs | https://www.ebi.ac.uk/emdb/EMD-25051 | Electron Microscopy Data Bank, EMD-25051 |
| Elferich J, Clark S, Ge J, Goehring A, Matsui A, Gouaux E | 2021 | Tomogram of mouse stereocilia containing PCDH15 molecules labeled with 39G7-AuNPs | https://www.ebi.ac.uk/emdb/EMD-25052 | Electron Microscopy Data Bank, EMD-25052 |
| Elferich J, Clark S, Ge J, Goehring A, Matsui A, Gouaux E | 2021 | Tomogram of mouse stereocilia containing PCDH15 molecules labeled with 39G7-AuNPs | https://www.ebi.ac.uk/emdb/EMD-25053 | Electron Microscopy Data Bank, EMD-25053 |

*Continued on next page*

*Continued*

| Author(s) | Year | Dataset title | Dataset URL | Database and Identifier |
|---|---|---|---|---|
| Elferich J, Clark S, Ge J, Goehring A, Matsui A, Gouaux E | 2021 | Tomogram of mouse stereocilia containing PCDH15 molecules labeled with 39G7-AuNPs | https://www.ebi.ac.uk/emdb/EMD-25054 | Electron Microscopy Data Bank, EMD-25054 |
| Elferich J, Clark S, Ge J, Goehring A, Matsui A, Gouaux E | 2021 | Tomogram of mouse stereocilia containing PCDH15 molecules labeled with 39G7-AuNPs | https://www.ebi.ac.uk/emdb/EMD-25055 | Electron Microscopy Data Bank, EMD-25055 |
| Elferich J, Clark S, Ge J, Goehring A, Matsui A, Gouaux E | 2021 | Tomogram of mouse stereocilia containing PCDH15 molecules labeled with 39G7-AuNPs | https://www.ebi.ac.uk/emdb/EMD-25056 | Electron Microscopy Data Bank, EMD-25056 |
| Elferich J, Clark S, Ge J, Goehring A, Matsui A, Gouaux E | 2021 | Tomogram of mouse stereocilia containing PCDH15 molecules labeled with 39G7-AuNPs | https://www.ebi.ac.uk/emdb/EMD-25057 | Electron Microscopy Data Bank, EMD-25057 |
| Elferich J, Clark S, Ge J, Goehring A, Matsui A, Gouaux E | 2021 | Tomogram of mouse stereocilia containing PCDH15 molecules labeled with 39G7-AuNPs | https://www.ebi.ac.uk/emdb/EMD-25058 | Electron Microscopy Data Bank, EMD-25058 |
| Elferich J, Clark S, Ge J, Goehring A, Matsui A, Gouaux E | 2021 | Tomogram of mouse stereocilia containing PCDH15 molecules labeled with 39G7-AuNPs | https://www.ebi.ac.uk/emdb/EMD-25059 | Electron Microscopy Data Bank, EMD-25059 |
| Elferich J, Clark S, Ge J, Goehring A, Matsui A, Gouaux E | 2021 | Tomogram of mouse stereocilia containing PCDH15 molecules labeled with 39G7-AuNPs | https://www.ebi.ac.uk/emdb/EMD-25060 | Electron Microscopy Data Bank, EMD-25060 |
| Elferich J, Clark S, Ge J, Goehring A, Matsui A, Gouaux E | 2021 | Tomogram of mouse stereocilia containing PCDH15 molecules labeled with 39G7-AuNPs | https://www.ebi.ac.uk/emdb/EMD-25061 | Electron Microscopy Data Bank, EMD-25061 |
| Elferich J, Clark S, Ge J, Goehring A, Matsui A, Gouaux E | 2021 | Molecular structure and conformation of stereocilia tip-links elucidated by cryo-electron tomography | https://www.ebi.ac.uk/empiar/EMPIAR-10820/ | Electron Microscopy Public Image Archive, EMPIAR-10820 |
| Elferich J, Clark S, Ge J, Goehring A, Matsui A, Gouaux E | 2021 | Cryo-electron tilt series of mouse stereocilia | https://www.ebi.ac.uk/empiar/EMPIAR-10898/ | Electron Microscopy Public Image Archive, EMPIAR-10898 |

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
