## [Editor Report]

The authors have done an extremely thorough job of responding to all of the concerns of the reviewers. The present manuscript will be a nice advance in the field of hearing research and is suitable for publication in *eLife*.

---

## [Decision Letter]

**Decision letter after peer review:**

Thank you for submitting your article "Molecular structure and conformation of stereocilia tip-links elucidated by cryo-electron tomography" for consideration by *eLife*. Your article has been reviewed by 2 peer reviewers, and the evaluation has been overseen by a Reviewing Editor and Richard Aldrich as the Senior Editor. The reviewers have opted to remain anonymous.

Essential revisions:

1) In a broad sense, any filamentous structure that bridges two stereociliary tips could be termed a "tip link." The conventional definition, though, restricts the term as defined by the authors on lines 21-22. Using the term more broadly risks erroneous conclusions: especially during murine development, there are clearly a wide variety of filamentous structures at stereociliary tips, nearly all of which disappear within the first two weeks of postnatal life. The term "tip link" should be stricken from instances in which the nature of the structures is not rigorously documented. This includes the submittal's title and sections such as that beginning at line 196, "Tomographic reconstructions of PCDH15-CDH23 heterotetramers." The authors have not confirmed the identities of such structures, and in particular have not demonstrated the presence of CDH23. The same holds for the section commencing on line 217, "An intact tip-link surrounded by non CDH23-bound PCDH15 molecules." The single structure in question is described as "most likely an intact tip-link," but that degree of proof does not inspire confidence in either the section title or indeed the title of the submittal itself.

2) The assertion on line 253-254 that "this work is the first look at native tip-links in situ" is unsubstantiated: the authors are staking an unjustified claim of priority. Although such an assertion should be supported by several well-documented structures, the actual evidence amounts to one "putative" tip link. Moreover, the structures are scarcely "native," given the violence of the bundle-blotting procedure and the acknowledged damage as a result; nor are they "in situ," which would imply their presence in a mouse's inner ear, or at least in living, functioning hair cells. The authors should describe what they have nicely demonstrated, the immunologically confirmed occurrence of PCDH15 molecules-many of them dimers-near stereociliary tips, and the association of those molecules with other as yet unidentified filamentous and lipidic structures.

3) The preparative techniques are sound and the tomographic images of generally high quality, as manifested for example by the smooth membrane profiles and nicely ordered actin microfilaments of the stereociliary cytoskeletons. Nevertheless, the interpretation of the data is shaky. Figure 3C gives striking evidence for a dimer of PCDH15. It is not at all clear, either in the figure or in the associated video, that the structure in panel (B) shows two labels; and that in panel (C) shows a dense blur, perhaps a cluster of gold particles. It is likewise unclear that there is any label bound in panel (D). In view of the ambiguous micrographs, a reader might well be skeptical about the ensuing statistical treatment.

---

## [Author Response]

Essential revisions:1) In a broad sense, any filamentous structure that bridges two stereociliary tips could be termed a "tip link." The conventional definition, though, restricts the term as defined by the authors on lines 21-22. Using the term more broadly risks erroneous conclusions: especially during murine development, there are clearly a wide variety of filamentous structures at stereociliary tips, nearly all of which disappear within the first two weeks of postnatal life. The term "tip link" should be stricken from instances in which the nature of the structures is not rigorously documented. This includes the submittal's title and sections such as that beginning at line 196, "Tomographic reconstructions of PCDH15-CDH23 heterotetramers." The authors have not confirmed the identities of such structures, and in particular have not demonstrated the presence of CDH23. The same holds for the section commencing on line 217, "An intact tip-link surrounded by non CDH23-bound PCDH15 molecules." The single structure in question is described as "most likely an intact tip-link," but that degree of proof does not inspire confidence in either the section title or indeed the title of the submittal itself.

We have removed references to tip links from the title and section headers and refer to PCDH15-complexes instead. For the section starting in line 217, we changed the section header to “A putative tip-link structure surrounded by non-complexes PCDH15 molecules”. We believe that this structure, even though it is a single observation, fulfills the most stringent definition of tip link, in that it is a filamentous connection from the tip of a stereocilium to the side of a taller stereocilium and contains PCDH15. We use “putative” to convey the remaining uncertainty in its assignment.

We have also removed references to CDH15/CDH23 heterotetramers and broadly talk about PCDH15 complexes. At some occasions we discuss that the complex partner is most likely CDH23, but qualify this by the use of “putative” or “probable”.

2) The assertion on line 253-254 that "this work is the first look at native tip-links in situ" is unsubstantiated: the authors are staking an unjustified claim of priority. Although such an assertion should be supported by several well-documented structures, the actual evidence amounts to one "putative" tip link. Moreover, the structures are scarcely "native," given the violence of the bundle-blotting procedure and the acknowledged damage as a result; nor are they "in situ," which would imply their presence in a mouse's inner ear, or at least in living, functioning hair cells. The authors should describe what they have nicely demonstrated, the immunologically confirmed occurrence of PCDH15 molecules-many of them dimers-near stereociliary tips, and the association of those molecules with other as yet unidentified filamentous and lipidic structures.

We have rephrased this assertion in line 253 with “This work is the first look at PCDH15 and associated protein and lipid membrane complexes in an unfixed, unstained environment, unambiguously demonstrating the presence of PCDH15 as a dimer”. We also use “near native” instead of native throughout the text to convey the nature of the specimen as derived from, but not residing in living hair cells. We have removed most uses of “in situ”, with the exception of the use of “in-situ cryo-electron microscopy”, which is the most widely used nomenclature to distinguish cryo-EM imaging of cellular samples from single-particle cryo-EM.

3) The preparative techniques are sound and the tomographic images of generally high quality, as manifested for example by the smooth membrane profiles and nicely ordered actin microfilaments of the stereociliary cytoskeletons. Nevertheless, the interpretation of the data is shaky. Figure 3C gives striking evidence for a dimer of PCDH15. It is not at all clear, either in the figure or in the associated video, that the structure in panel (B) shows two labels; and that in panel (C) shows a dense blur, perhaps a cluster of gold particles. It is likewise unclear that there is any label bound in panel (D). In view of the ambiguous micrographs, a reader might well be skeptical about the ensuing statistical treatment.

While the interpretation of tomograms can be subjective, as pointed out here and in other comments, the presence of two (Figure 3A+B) or one (Figure 3C+D) AuNPs is unambiguous. However, it can be difficult to convey this by 2D slices through a 3D density, especially when slices that would show 2 AuNPs are affected by the ‘missing-wedge’ problem of electron-tomography. We have updated Figure 3 with consistent slices from a “top” and “side”-view which, together with an updated figure legend, will make this clearer. We have also updated Videos 2-5 to make clearer how the annotations are derived from the density.